# Association of Serum Magnesium with Insulin Resistance and Type 2 Diabetes among Adults in China

**DOI:** 10.3390/nu14091799

**Published:** 2022-04-25

**Authors:** Weiyi Li, Yingying Jiao, Liusen Wang, Shaoshunzi Wang, Lixin Hao, Zhihong Wang, Huijun Wang, Bing Zhang, Gangqiang Ding, Hongru Jiang

**Affiliations:** 1Office of National Nutrition Plan, National Institute for Nutrition and Health, Chinese Center for Disease Control and Prevention, 27 Nanwei Road, Beijing 100050, China; liwy@ninh.chinacdc.cn (W.L.); jyy2227085940@163.com (Y.J.); wangls@ninh.chinacdc.cn (L.W.); wangssz@ninh.chinacdc.cn (S.W.); haolx@ninh.chinacdc.cn (L.H.); wangzh@ninh.chinacdc.cn (Z.W.); wanghj@ninh.chinacdc.cn (H.W.); zhangbing@chinacdc.cn (B.Z.); dinggq@chinacdc.cn (G.D.); 2Key Laboratory of Trace Elements and Nutrition, National Health Commission, Beijing 100050, China

**Keywords:** serum magnesium, insulin resistance, diabetes, China

## Abstract

Magnesium is an essential mineral for the human body and a cofactor or activator for more than 300 enzymatic reactions, including blood glucose control and insulin release. Diabetes is a well-known global burden of disease with increasing global prevalence. In China, the prevalence of diabetes in adults is higher than the global average. Evidence shows that magnesium is a predictor of insulin resistance and diabetes. However, the majority of studies focus on dietary magnesium instead of serum magnesium concentration. We study the correlation of serum magnesium levels with insulin resistance and Type 2 diabetes. In this prospective cohort study, we included 5044 participants aged 18 years and older without insulin resistance (IR) and diabetes at the baseline from China Health and Nutrition Survey (CHNS). A fasting blood sample was taken for the measurement of both types of magnesium, fasting blood glucose, hemoglobin A1c (HbA1c), and fasting insulin. The homeostatic model (HOMA-IR) was calculated. Demographic characteristics of participants, and risk factors such as intensity of physical activities, smoking status, drinking habit, and anthropometric information were recorded. IR was defined as HOMA-IR ≥ 2.5, and Type 2 diabetes mellitus was defined as fasting plasma glucose ≥ 7.0 mmol/L or HbA1c ≥ 6.5%, or a self-reported diagnosis or treatment of diabetes. A total of 1331 incident insulin resistance events and 429 incident diabetic events were recorded during an average follow-up of 5.8 years. The serum magnesium concentration was categorized into quintiles. After adjusting for relevant covariates, the third quintile of serum magnesium (0.89–0.93 mmol/L) was correlated with 29% lower risk of incident insulin resistance (hazard ratio = 0.71, 95% CI 0.58, 0.86) and with a lower risk of Type 2 diabetes. Multivariable-adjusted hazard ratios (95% confidence intervals) for insulin resistance were compared with the lowest quintile of serum magnesium (<0.85). We found similar results when evaluating serum magnesium as a continuous measure. Restricted cubic spline (RCS) curves showed a nonlinear dose–response correlation in both serum magnesium levels and insulin resistance, and in serum magnesium levels and Type 2 diabetes. Lower serum magnesium concentration was associated with a higher risk of insulin resistance and diabetes.

## 1. Introduction

Diabetes is a global burden of disease that is globally growing at a remarkable rate. In 2017, the global prevalence of Type 2 diabetes was 6059 cases per 100,000, with roughly 462 million individuals being affected by Type 2 diabetes, equivalent to 6.28% of the global population [1]. In China, chronic disease surveillance data showed that the prevalence of diabetes mellitus in adults was higher than the global average in 2018, up to 12.8%, while the awareness rate of diabetes was only 36.7% [2]. Globally, the prevalence rate continues to rise regardless of socioeconomic factors, and the global prevalence of Type 2 diabetes will increase to 7079 in every 100,000 people by 2030 [1]. Insulin is a hormone that is produced by pancreatic β-cells; the primary function of insulin is to maintain glucose homeostasis in tissue such as skeletal muscle, adipose tissue, and the liver through promoting glucose transport and inhibiting glucose production [3,4]. β-cells failing to compensate for peripheral insulin resistance (IR) result in the development of Type 2 diabetes, and the presence of peripheral insulin resistance in tissue is a characteristic of diabetes [4].

Magnesium is one of the most abundant intracellular divalent cations, second only to potassium, and the fourth most plentiful cation in the human body [5]. Magnesium balance in the human body is controlled by the dynamic interaction among intestinal absorption, renal resorption and excretion, and magnesium conversion in bone tissue. About 30% of magnesium enters the blood through paracellular and transcellular pathways [6]. Magnesium is vital to enzymatic reactions as a cofactor or activator for hundreds of enzymes [7]. In fact, magnesium participates in more than 300 enzymatic reactions in the human body, and regulates various biochemical reactions, including blood glucose control. An animal study showed that magnesium controls glucose tolerance through the suppression of the gluconeogenesis pathway and glucagon receptor gene expression, and the stimulation of glucose transporter 4 (Glut4) gene expression and GLUT4 protein translocation in the liver and gastrocnemius muscle [8]. Although the molecular mechanisms of magnesium-induced insulin resistance have not been fully revealed, it is clear that magnesium plays an essential role in inducing insulin release [5]. The activity of protein tyrosine kinases is dependent on the concentration of intracellular magnesium; consequently, lower magnesium concentration can inhibit insulin action, increasing insulin tolerance [9]. A recent study demonstrated that magnesium supplementation improves insulin sensitivity by affecting β-arrestin-2 gene expression and inhibiting lipid peroxidation [10].

Magnesium is associated with many chronic and inflammatory diseases or conditions, such as Alzheimer’s disease, asthma and respiratory insufficiency, psychiatric disorders, insulin resistance, Type 2 diabetes, hypertension, cardiovascular disease (e.g., stroke), migraines, osteoporosis, and cancer [11]. In epidemiological studies, magnesium has attracted wide attention due to its role in the prevention of diabetes. Diabetes is usually accompanied by changes in magnesium status. According to a Canadian study, patients with diabetes mellitus had lower serum magnesium levels (between 0.04 and 0.07 mmol/L) compared with nondiabetic participants, and insulin homeostatic model assessment was negatively associated with serum magnesium levels [12].

In this study, we analyze the association of serum magnesium with IR and Type 2 diabetes mellitus within a large longitudinal cohort.

## 2. Subjects and Methods

### 2.1. Study Population

Research data of this study are based on the China Health and Nutrition Survey (CHNS), an ongoing multipurpose longitudinal survey to assess secular trends in health and nutritional status, and to research the etiology of nutritional diseases in the Chinese population. The CHNS was established in 1989, covering more than 20,000 subjects from 15 provinces by 2015 using a multistage random-cluster sampling process to draw the survey data from each selected province in China [13]. Our analysis used survey data from three rounds (2009, 2015, 2018) of the CHNS, since fasting blood samples were collected in the CHNS in 2009, 2015, and 2018. Eligible participants were 18 years of age and older with reasonable BMI (BMI ≥ 14 and ≤45), not pregnant or lactating, had completed the fasting blood collection for biochemical measurements, and had complete questionnaire records. Our analysis excluded individuals with diabetes or insulin resistance at baseline (*n* = 6690) and those with only 1 round follow-up (*n* = 5978). A total of 5044 participants were included in the final analysis. All participants who enrolled in the study signed written informed consent to participate in the survey.

### 2.2. Exposure and Outcome Assessment

Participants had fasted for 8 to 12 h at the time of blood sample collection. Standard operating procedures regarding blood-sample collection, processing, and storage were followed. Fasting blood samples were analyzed at the laboratory of the National Institute for Nutrition and Health (NINH), China CDC. The primary predictor was serum magnesium concentration, which was measured in mmol/L by the xylidyl-blue colorimetric method. Fasting glucose was measured with the glucose oxidase-phenol aminophenazone method, hemoglobin A1c (HbA1c) was measured with an automated glycohemoglobin analyzer, and insulin was measured with ECL [14,15].

The diagnostic criterion for Type 2 diabetes was based on the guideline for the prevention and treatment of Type 2 diabetes mellitus in China (2020 edition) [16] and the World Health Organization criteria for diabetes mellitus (fasting glucose ≥ 7.0 mmol/L or HbA1c ≥ 6.5%), or a self-reported diagnosis of diabetes and treatment with antidiabetic pharmacotherapy. The homeostatic model (HOMA-IR) was used as the surrogate measure of IR, calculated as (fasting glucose (mmol/L) × fasting insulin (μU/mL))/22.5 [17], and IR was defined as HOMA-IR ≥ 2.5 [18].

### 2.3. Covariate Assessment

A well-designed general questionnaire was used to collect participants’ sociodemographic, lifestyle, and anthropometric information (including age, sex, education (low: primary school and below; medium: middle or high school; high: college and above), residential area (urban or rural), annual household income, smoking status, drinking habit, and physical activity) during the inhouse interview conducted by well-trained health professionals, following the reference protocol written by NINH. Body mass index was calculated by height and weight measurements (BMI, kg/m^2^) and categorized into three levels (<18.5 kg/m^2^, 18.5–23.9 kg/m^2^ and ≥24.0 kg/m^2^). Household income was categorized into tritiles (low, medium, and high). Physical activity was calculated by metabolic-equivalent hours/week (MET-h/week), which categorized by the intensity of physical activity assignments [19] into tertiles (low, medium, and high). Age was categorized into three categories (18–49, 50–64, and 65+) for the adjustment of Cox multivariate proportional hazard regression models.

### 2.4. Statistical Analysis

Descriptive statistics of participants at baseline across serum magnesium quintiles are presented; continuous variables with normal distribution are presented in means ± standard deviation, continuous variables with non-normal distribution are presented in media (p25, p75), and categorical variables are presented in percentages. Analysis of variance was used for comparison of groups with continuous baseline variables, including sociodemographic, lifestyle, and anthropometric variables across quintile groups of serum magnesium; the Mantel–Haenszel chi-squared test was used to compare categorical baseline variables within quintile groups.

Associations between levels of serum magnesium and the risk of insulin resistance, and Type 2 diabetes mellitus were estimated by Cox multivariate proportional hazards regression models with age as the underlying time scale (participants enter the risk set at the baseline age and exit at the censoring or event age). Model 1 was adjusted for age (three categories: 18–49, 50–64, and 65+), sex, education, residential area, and household income. Model 2 was adjusted as Model 1 plus lifestyle risk factors (physical activity, smoking status, and drinking habit). Model 3 was further adjusted for BMI. Hazard ratios (HRs) with 95% confidence intervals (CIs) were calculated accordingly.

To explore the nonlinear dose–response correlation between levels of serum magnesium and all endpoints, restricted cubic spline (RCS) curves based on Cox multivariate proportional hazards models [20] with five knots were produced.

To test the robustness of our findings, sensitivity analysis was performed. We repeated our analysis on the basis of HOMA-IR ≥ 2.0 [21] as an indicator of insulin resistance to study whether different diagnostic criteria of IR could have influenced our effect estimates.

All statistical analyses of this study were performed with SAS software package version 9.4 (SAS Institute, Inc., Cary, NC, USA) and R software version 4.1.0 (The R Foundation for Statistical Computing). A *p*-value of less than 0.05 was considered to be statistically significant.

## 3. Results

### 3.1. Baseline Characteristics

As presented in Table 1, among the 5044 participants at baseline, those in the lower serum magnesium quintiles were more likely to be female, current smokers, and had a drink habit. Those in the higher serum magnesium quintiles were more likely to have a higher household income and lower intensity of physical activity. Fasting blood glucose and fasting insulin were lower in the lower quintiles of serum magnesium. Most participants fell within the recommended serum magnesium levels (0.75–0.95 mmol/L) [22].

### 3.2. Association of Serum Magnesium with Insulin Resistance

During the average follow-up of 5.8 years, 1331 participants were identified as insulin resistance cases. Table 2 presents the association between quintiles of serum magnesium and insulin resistance (HOMA-IR ≥ 2.5). After adjusting for age, sex, education, residential area, and annual household income (Table 2), HRs (95% CI) of IR for each quintile compared to the first quintile of serum magnesium were: Q2: 0.93 (0.78–1.11); Q3: 0.73 (0.61–0.88) Q4: 0.84 (0.71–1.00); and Q5: 0.97 (0.82–1.16); p-trend = 0.41. The association remained statistically significant in Q3 and Q4 compared with the lowest quintile of serum magnesium after further adjustment for lifestyle covariates (including physical activity, smoking status, drinking habit) and BMI with p-trends of 0.34 and 0.19, respectively (Table 2).

### 3.3. Association of Serum Magnesium with Type 2 Diabetes

Among the 5044 participants without Type 2 Diabetes and with normal fasting blood glucose at baseline, 429 cases of Type 2 diabetes were identified during an average of 5.8 years of follow-up. After adjusting for age, sex, education, residential area, and annual household income (Table 3), HRs (95% CI) of Type 2 diabetes for each quintile compared to the first quintile of serum magnesium were: Q2: 0.76 (0.56–1.05); Q3: 0.68 (0.49–0.95) Q4: 0.75 (0.55–1.02); and Q5: 0.90 (0.67–1.22); p-trend = 0.82. The association remained statistically significant in Q3 compared with the lowest quintile of serum magnesium group after further adjustment for lifestyle covariates (including physical activity, smoking status, drinking habit), with a p-trend of 0.82 (Table 3). However, the association was statistically significant not only in Q3 compared with the lowest quintile of serum magnesium after fully adjustment, but also in Q4 after adjustment for sociodemographic, lifestyle, and BMI characteristics. The multivariable-adjusted HRs for Type 2 diabetes comparing Q3 and Q4 to Q1 of serum magnesium were 0.68 (95% CI, 0.49–0.96) and 0.69 (95% CI, 0.50–0.95), respectively.

### 3.4. Dose–Response Relationship between Serum Magnesium and Insulin Resistance, and Type 2 Diabetes

Figure 1 shows the result of dose–response relationship between serum magnesium levels and insulin resistance (HOMA-IR ≥ 2.5) by RCS model. The hazard ratio of IR and serum magnesium displayed a nonlinear relationship (nonlinear *p* < 0.001) while setting the 5th percentile of serum magnesium level (0.78 mmol/L) as a reference. When the serum magnesium level was less than 0.82 mmol/L, the hazard ratio of IR progressively rose with increasing serum magnesium. When the serum magnesium level reached around 0.82 mmol/L, the risk of IR decreased with the increase in serum magnesium levels. However, when the serum magnesium was greater than approximately 0.93 mmol/L, the risk of IR rapidly rose.

The result of the dose–response relationship between serum magnesium levels and Type 2 diabetes mellitus in the RCS model showed that the hazard ratio of Type 2 diabetes and serum magnesium displayed a similar nonlinear relationship (nonlinear *p* < 0.001) comparing the relationship of IR and serum magnesium. V-shaped correlation was found when setting the 5th percentile of serum magnesium level (0.78 mmol/L) as a reference. When serum magnesium level was less than approximately 0.93 mmol/L, the risk of diabetes decreased with increasing serum magnesium levels. However, when the serum magnesium level reached 0.93 mmol/L, the risk of diabetes increased with increasing serum magnesium (Figure 2).

### 3.5. Sensitivity Analyses

Sensitivity analyses showed a similar relationship between serum magnesium with IR and diabetes mellitus as that in our main analyses. Multivariable-adjusted HRs for IR comparing Q3 and Q4 to the lowest quintile of serum magnesium level were 0.71 (95% CI, 0.57–0.89) and 0.67 (95% CI, 0.53–0.83), respectively. The association between serum magnesium levels and Type 2 diabetes mellitus remained statistically significant in Q4 compared with the lowest quintile of serum magnesium level after fully adjusting for sociodemographic, lifestyle, and BMI characteristics (HR 0.70 (95% CI, 0.50–0.98)).

## 4. Discussion

Findings from this prospective cohort study among 5044 participants with an average follow-up of 5.8 years showed that participants with serum magnesium level between 0.89 and 0.93 had the lowest risk of IR and Type 2 diabetes compared with those at the lowest quintile of serum magnesium level. In the present analysis, we saw a nonlinear dose–response association between serum magnesium levels, and IR and diabetes. In fact, we found U-shaped correlation between serum magnesium levels, and the risk of IR and Type 2 diabetes, with low and high levels associated with increased risk. Dose–response analyses showed that participants with a serum magnesium level around 0.93 mmol/L had the lowest risk of IR (HOMA-IR ≥ 2.5) and Type 2 diabetes mellitus. The prevalence of IR and Type 2 diabetes of the study population was 26.39% and 8.51%, respectively, and the prevalence of Type 2 diabetes mellitus in our study was slightly greater than the estimated level (7.5%) investigated on CHNS 2009 [23].

Some studies suggested significantly negative correlation between serum magnesium and insulin resistance [24,25,26]; however, there were also some studies showing that the relationship is insignificant [27,28]. One study by Akter et al. found that serum magnesium levels were inversely correlated with HOMA-IR and HbA_1c_, but the association was attenuated in healthy subjects [27].

Previous studies found an inverse association between lower serum magnesium levels and risk of diabetes mellitus [26,29,30,31,32,33], and some studies showed that lower serum magnesium levels were positively associated with poor glycemic control [34,35,36,37]. In a prospective Rotterdam population-based cohort study in people aged 45 and over, among 8555 participants, the risk of prediabetes and diabetes increased with decreasing serum magnesium levels, while insulin resistance was a mediating factor [26]. Several studies showed that Type 2 diabetic or prediabetic patients were more likely to have lower serum magnesium levels [35,38,39,40,41]. Secondary analysis regarding a double-blind placebo-controlled randomized clinical trial of vitamin D supplementation found that serum magnesium level was significantly lower in patients with Type 2 diabetes [38]. The meta-analysis of 13,455 subjects (2979 prediabetic patients and 10,476 healthy controls) revealed that serum magnesium level were significant lower in prediabetic subjects than those of healthy controls [39]. Moreover, a recent case–control study in Morocco confirmed that serum magnesium levels were positively correlated with glycemic control; among 170 patients with Type 2 diabetes, HbA1c was significant higher in patients with lower magnesium levels [35]. Those studies strongly showed that hypomagnesemia is not an important feature of diabetes, but a valuable factor to predict diabetes. A prospective study suggested serum magnesium concentrations ≥0.82 mmol/L as an evidence-based reference interval of serum magnesium concentrations for preventing noncommunicable chronic diseases such as hypertension, stroke, and Type 2 diabetes [42]. Genetics seems to be a vital risk factor for Type 2 diabetes, and serum magnesium levels may modify the risk of diabetes through several magnesium-regulating genes such as TRPM6, CLDN19, SLC41A2, CNNM2, and FXYD2 [26]. The mechanism of how CNNM2 and FXYD2 mediated the serum magnesium levels has still not been revealed [43]. Both magnesium and zinc are vital micronutrients in glucose homeostasis [44,45], and deficiency in magnesium and zinc could lead to complications of diabetes mellitus [46,47]. Several meta-analyses indicated that zinc supplementation benefits glucose control and diabetes mellitus management [48,49]. Hamedifard et al. conducted a trial on Type 2 diabetes patients with coronary heart disease, and found that the levels of insulin and fasting blood glucose were significantly improved in participants taking zinc and magnesium supplements [45].

The strengths of our study include the large sample size, and the stability and detailed characterization of the study population. CHNS is a longitudinal cohort study tracking data in two generations that provides study populations with heterogeneity. Furthermore, each round of CHNS strictly followed a protocol with the same survey design and questionnaire model to ensure the comparability of data across different rounds. This is the first prospective cohort that evaluated the relationship between serum magnesium, and IR and Type 2 diabetes among healthy adults in China. In addition, we chose data with at least two rounds of follow-up to obtain more reliable information in order to examine the association of serum magnesium levels with IR and Type 2 diabetes mellitus. Furthermore, we used RCS analysis to explore the dose–response correlation between continuous serum magnesium levels, and IR and Type 2 diabetes.

A potential limitation of our study is that we excluded participants on the basis of missing or invalid values on other variables of interest, which may have significant differences in sociodemographic, lifestyle, and anthropometric variables compared with the remaining analytical subjects. The quantile division of continuous variables may have caused misclassification bias, and there were information and batch biases in our study caused by the multiround cohort study design and data collection. We also could not rule out potential confounding factors caused by the inability to obtain data from the current survey such as family history of diabetes and treatment methods or by inaccurately measured factors. In our study, the micronutrient amount taken from nutrient supplements and fortified food was not evaluated. Moreover, we could not estimate the daily water intake in terms of Mg^2+^ during the follow-up period.

## 5. Conclusions

Lower serum magnesium was associated with IR and Type 2 diabetes in participants with normal serum magnesium levels. Nonlinear dose–response relationships were found in both serum magnesium and IR, and in serum magnesium and Type 2 diabetes. Potential reasons for the U-shaped association between serum magnesium levels, and the risk of IR and diabetes warrant further discussion.

## Figures and Tables

**Figure 1 nutrients-14-01799-f001:**
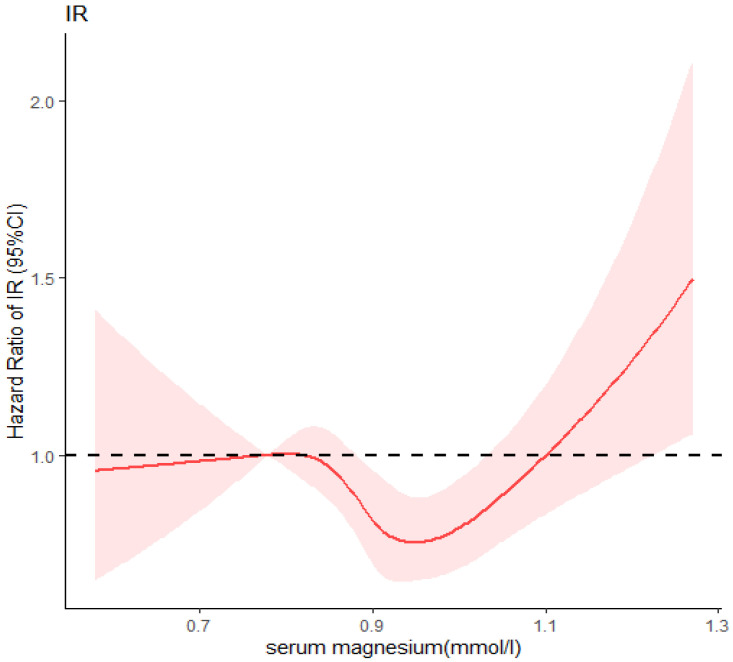
Restricted cubic spline of association between serum magnesium and IR (HOMA-IR ≥ 2.5).

**Figure 2 nutrients-14-01799-f002:**
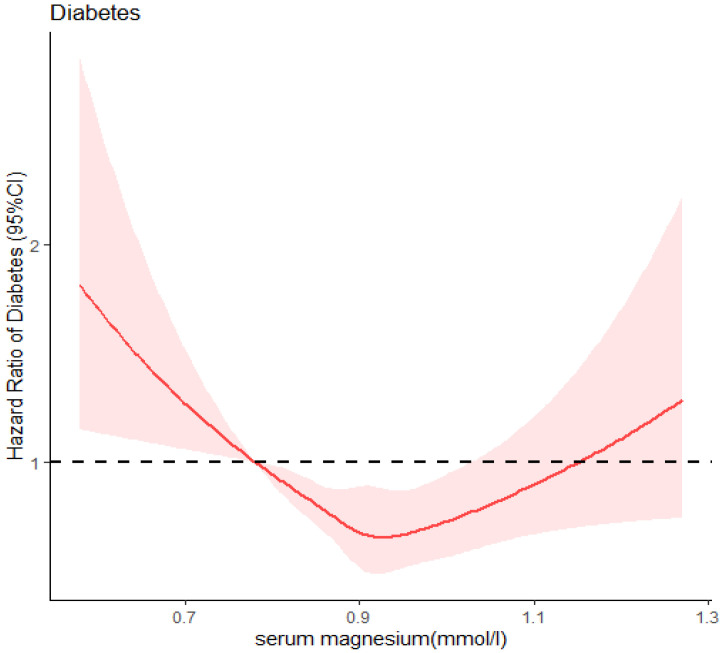
Restricted cubic spline of association between serum magnesium and Type 2 diabetes.

**Table 1 nutrients-14-01799-t001:** Characteristics of participants by serum magnesium quintile.

Characteristic	Quintile of Serum Magnesium (mmol/L)					*p*-Value
	Q1 (<0.85)	Q2 (0.85–0.89)	Q3 (0.89–0.93)	Q4 (0.93–0.98)	Q5 (≥0.98)	
No. of subjects	948	1110	906	1082	1008	
Age at baseline (mean ± SD) ^a^	49.7 ± 13.2	51.5 ± 13.6	52.0 ± 13.3	52.4 ± 13.0	53.1 ± 13.0	<0.001
Age (*n*, %) ^b^						
18–49	486 (51.27)	506 (46.00)	387 (42.72)	444 (41.04)	378 (37.50)	<0.001
50–64	343 (36.18)	410 (37.27)	364 (40.18)	460 (42.51)	455 (45.14)	
65+	119 (12.55)	184 (16.73)	155 (17.11)	178 (16.45)	175 (17.36)	
Female (*n*, %) ^b^	577 (60.86)	623 (56.64)	523 (57.73)	567 (52.40)	471 (46.73)	<0.001
Education (*n*, %) ^b^						
low	362 (38.19)	389 (35.36)	364 (40.18)	401 (37.06)	356 (35.32)	0.323
Medium	436 (45.99)	521 (47.36)	408 (45.03)	506 (46.77)	468 (46.43)	
high	150 (15.82)	190 (17.27)	134 (14.79)	175 (16.17)	184 (18.25)	
Urban (*n*, %) ^b^	310 (32.70)	410 (37.27)	316 (34.88)	350 (32.35)	328 (32.54)	0.073
Household income (*n*, %) ^b^						
Low	318 (33.54)	369 (33.55)	331 (36.53)	370 (34.20)	293 (29.07)	<0.001
Medium	345 (36.39)	359 (32.64)	311 (34.33)	352 (32.53)	314 (31.15)	
High	285 (30.06)	372 (33.82)	264 (29.14)	360 (33.27)	401 (39.78)	
Never smoked (*n*, %) ^b^	239 (25.21)	288 (26.18)	261 (28.81)	339 (31.33)	333 (33.04)	<0.001
Never drank alcohol (*n*, %) ^b^	262 (27.64)	340 (30.91)	278 (30.68)	357 (32.99)	357 (35.42)	0.004
Physical activity (*n*, %) ^b^						
Low	282 (29.75)	373 (33.91)	285 (31.46)	370 (34.20)	371 (36.81)	0.019
Medium	343 (36.18)	372 (33.82)	299 (33.00)	334 (30.87)	330 (32.74)	
High	323 (34.07)	355 (32.27)	322 (35.54)	378 (34.94)	307 (30.46)	
BMI (kg/m^2^) (*n*, %) ^b^						
<18.5	48 (5.06)	64 (5.82)	56 (6.18)	64 (5.91)	52 (5.16)	0.159
18.5–23.9	540 (56.96)	593 (53.91)	501 (55.30)	556 (51.39)	518 (51.39)	
≥24.0	360 (37.97)	443 (40.27)	349 (38.52)	462 (42.70)	438 (43.45)	
Fasting blood glucose (mmol/L) ^a^	4.89 (4.55, 5.28)	4.99 (4.62, 5.35)	4.97 (4.63, 5.36)	5.00 (4.60, 5.39)	5.04 (4.62, 5.46)	<0.001
Fasting insulin (µU/mL) ^a^	6.85 (4.97, 8.79)	7.74 (5.77, 9.33)	7.55 (5.79, 9.21)	7.88 (6.26, 9.35)	7.41 (5.72, 9.08)	<0.001
HOMA-IR ^a^	1.47 (1.02, 1.95)	1.73 (1.25, 2.05)	1.67 (1.25, 2.05)	1.71 (1.33, 2.08)	1.65 (1.22, 2.01)	<0.001

BMI, body mass index; HOMA-IR, homeostatic model assessment-insulin resistance. Values are media (p25, p75) for continuous variables, and percentages for categorical characteristics. ^a^ Kruskal–Wallis test was used to calculated *p*-value for non-normal distribution continuous variables; ^b^ Mantel–Haenszel χ^2^ was used to calculated *p*-value for categorical variables.

**Table 2 nutrients-14-01799-t002:** Hazard ratio (HR) and 95% confidence interval (CI) for association between serum magnesium quintiles and insulin resistance risk.

	Quintile of Serum Magnesium					*p*-Trend
	Q1 (<0.85)	Q2 (0.85–0.89)	Q3 (0.89–0.93)	Q4 (0.93–0.98)	Q5 (≥0.98)	
Serum magnesium (mmol/L) medium (p25, p75)	0.81 (0.78, 0.83)	0.87 (0.86, 0.89)	0.91 (0.90, 0.92)	0.95 (0.94, 0.96)	1.02 (1.00, 1.05)	
Model 1	1.00	0.93 (0.78, 1.11)	0.73 (0.61, 0.88) *	0.84 (0.71, 1.00)	0.97 (0.82, 1.16)	0.41
Model 2	1.00	0.92 (0.77, 1.09)	0.72 (0.60, 0.87) *	0.83 (0.69, 0.99) *	0.96 (0.80, 1.14)	0.34
Model 3	1.00	0.90 (0.75, 1.08)	0.71 (0.58, 0.86) *	0.76 (0.63, 0.91) *	0.92 (0.77, 1.11)	0.19

* *p* < 0.05. Cox’s proportional hazard models were used to estimate HR. Model 1 was adjusted for age (three categories: 18–49, 50–64, and 65+), sex, education (primary school and below, middle or high school, and at least college), residential area (urban or rural), and household income (low, medium, high). Model 2 was adjusted as Model 1 plus lifestyle risk factors (physical activity (low, medium, high), smoking status (yes or no), and drinking habit (yes or no)). Model 3 was further adjusted for BMI (<18.5, 18.5–23.9, and ≥24.0 kg/m^2^).

**Table 3 nutrients-14-01799-t003:** Hazard Ratio (HR) and 95% confidence interval (CI) for association between serum magnesium quintiles and Type 2 diabetes risk.

	Quintile of Serum Magnesium					*p*-Trend
	Q1 (<0.85)	Q2 (0.85–0.89)	Q3 (0.89–0.93)	Q4 (0.93–0.98)	Q5 (≥0.98)	
Serum magnesium (mmol/L) medium (p25, p75)	0.81 (0.78, 0.83)	0.87 (0.86, 0.89)	0.91 (0.90, 0.92)	0.95 (0.94, 0.96)	1.02 (1.00, 1.05)	
Model 1	1.00	0.76 (0.56, 1.05)	0.68 (0.49, 0.95) *	0.75 (0.55, 1.02)	0.84 (0.62, 1.15)	0.82
Model 2	1.00	0.81 (0.59, 1.12)	0.69 (0.50, 0.97) *	0.75 (0.55, 1.03)	0.88 (0.63, 1.22)	0.82
Model 3	1.00	0.78 (0.56, 1.08)	0.68 (0.49, 0.96) *	0.69 (0.50, 0.95) *	0.87 (0.63, 1.20)	0.50

* *p* < 0.05. Cox’s proportional hazard models were used to estimate HR. Model 1 was adjusted for age (three categories: 18–49, 50–64, and 65+), sex, education (primary school and below, middle or high school, and at least college), residential area (urban or rural), and household income (low, medium, high). Model 2 was adjusted as Model 1 plus lifestyle risk factors (physical activity (low, medium, high), smoking status (yes or no), and drinking habit (yes or no)). Model 3 was further adjusted for BMI (<18.5, 18.5–23.9, and ≥24.0 kg/m^2^).

## Data Availability

Data sharing is not applicable to this article.

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
