# Peer review of "Association of Serum Magnesium with Insulin Resistance and Type 2 Diabetes among Adults in China"

_nutrients, 2022, doi:10.3390/nu14091799_

Round 1
Reviewer 1 Report
It is a very practical and well-planned study. IT is distinguished by a large sample size (5044) and a fairly long observation period (5,8 years).
What's innovative is that the authors showed that partisipants with serum magnesium level between 0,89 to 0,93 mmol/L had lowest risk of IR and type 2 diabetes. Moreover, they indicated a V-shaped corrltation with serum magnesium level and IR and diabetes. This is very important from a clinical point of view, as both lower than 0,82 mmol/L and higher levels of magnesium than 0,93 mol/|L were associated with a higher risk of IR and diabetes.
I am only asking for information whether the participants supplemented the magnesium supplement in any way?
Why do the authors use the phrase "dose-response"- isn't it the concentration of magnesium in the serum?
Reference standards for magnesium levels should be given.
Author Response
Point 1: I am only asking for information whether the participants supplemented the magnesium supplement in any way?
Response 1: Unfortunately, in our dataset there is no information on whether the participants took any supplement. We will add this into discussion section as one of limitations.
Point 2: Why do the authors use the phrase "dose-response"- isn't it the concentration of magnesium in the serum? Reference standards for magnesium levels should be given
Response 2: According to the Endocrine Disruption and Human Health (Second Edition), 2022, the term “dose response” simply can be refers to the relationship between the applied dose/concentration (the amount of a substance administered, purposely or inadvertently, to cultured cells, an animal, or a person) and the effect that is observed. One of the objective is to investigate the dose-response relationships of serum magnesium levels with risk of IR and type 2 diabetes mellitus. From an epidemiological point of view that it is ok to use the term “dose-response” to explain the relationship between serum magnesium levels and risk of certain outcomes.
We use the 5th percentile of serum magnesium level (0.78 mmol/L) as a reference in RCS model for dose response analyses. The normal serum magnesium concentrations range between 0.75 and 0.95 millimoles (mmol)/L. Hypomagnesemia is defined as a serum magnesium level less than 0.75 mmol/L.
Reviewer 2 Report
The manuscript entitled "Association of serum magnesium with insulin resistance and type 2 diabetes among adults in China" describes the relationship between serum magnesium and type 2 diabates. The work is designed and presented. However, authors should include the role of zinc in addition to magnesium in the introduction as both concentrations decreases with type 2 diabetes.
Authors should also include and discuss some references where both zinc and magnesium are used as a supplement to reduce the risk.
Lipids in Health and Disease Volume 19, Article number: 112 (2020)
Also, the effect of medication such as metformin on the level of magnesium in the serum.
European Journal of Molecular & Clinical Medicine, 2020, Volume 7, Issue 11, Pages 609-616
Author Response
Point 1: Authors should also include and discuss some references where both zinc and magnesium are used as a supplement to reduce the risk.
Lipids in Health and Disease Volume 19, Article number: 112 (2020)
Also, the effect of medication such as metformin on the level of magnesium in the serum.
European Journal of Molecular & Clinical Medicine, 2020, Volume 7, Issue 11, Pages 609-616
Response 1: Yes, we would add some references about the effects of combined zinc and magnesium supplementation on metabolic status and the effect of medication in the discussion section. Unfortunately, the current survey did not collect the information regarding serum zinc level and the specific categories of medication taken by participants.
Changes in discussion section as “Both magnesium and zinc are vital micronutrients in glucose homeostasis[44,45], and deficiency in magnesium and zinc could lead to develop complications of diabetes mellitus[46,47]. Several meta-analyses indicated that zinc supplementation benefit on glucose control and diabetes mellitus managing[48,49]. Hamedifard et al. conducted a trial of type 2 diabetes patients with coronary heart disease, and found that the levels of insulin and fasting blood glucose were significant improved in participants taking zinc and magnesium supplements[45].”
"Besides, we cannot rule out the potential confounding factors caused by unable to obtain from the current survey such as family history of diabetes, treatment methods, or caused by inaccurately measured factors. In our study, the amount of micronutrients taken from nutrient supplements and fortified food was not evaluated. "
We have also added some references in our study:
Petroni, M.L.; Brodosi, L.; Marchignoli, F.; Sasdelli, A.S.; Caraceni, P.; Marchesini, G.; Ravaioli, F. Nutrition in Patients with Type 2 Diabetes: Present Knowledge and Remaining Challenges. Nutrients 2021, 13, doi:10.3390/nu13082748.
Hamedifard, Z.; Farrokhian, A.; Reiner, Ž.; Bahmani, F.; Asemi, Z.; Ghotbi, M.; Taghizadeh, M. The effects of combined magnesium and zinc supplementation on metabolic status in patients with type 2 diabetes mellitus and coronary heart disease. Lipids in health and disease 2020, 19, 112, doi:10.1186/s12944-020-01298-4.
Feng, J.; Wang, H.; Jing, Z.; Wang, Y.; Wang, W.; Jiang, Y.; Sun, W. Relationships of the Trace Elements Zinc and Magnesium With Diabetic Nephropathy-Associated Renal Functional Damage in Patients With Type 2 Diabetes Mellitus. Frontiers in medicine 2021, 8, 626909, doi:10.3389/fmed.2021.626909.
Kuzhandai Velu VENGATAPATHY , A.V.C., Shaik Anwar Hussain K , Kalai Selvi RAJENDIRAN , Monisha MOHAN, Lenin MUNISAMMY ,. Serum Zinc and Magnesium Levels in Type 2 Diabetes Mellitus Patients on Metformin Therapy. European Journal of Molecular & Clinical Medicine 2020, 7, 609-616.
Jafarnejad, S.; Mahboobi, S.; McFarland, L.V.; Taghizadeh, M.; Rahimi, F. Meta-Analysis: Effects of Zinc Supplementation Alone or with Multi-Nutrients, on Glucose Control and Lipid Levels in Patients with Type 2 Diabetes. Preventive nutrition and food science 2019, 24, 8-23, doi:10.3746/pnf.2019.24.1.8.
Wang, X.; Wu, W.; Zheng, W.; Fang, X.; Chen, L.; Rink, L.; Min, J.; Wang, F. Zinc supplementation improves glycemic control for diabetes prevention and management: a systematic review and meta-analysis of randomized controlled trials. The American journal of clinical nutrition 2019, 110, 76-90, doi:10.1093/ajcn/nqz041